Linear B-cell epitope prediction for SARS and COVID-19 vaccine design: Integrating balanced ensemble learning models and resampling strategies

Gurcan Fatih fgurcan@ktu.edu.tr
Department of Management Information Systems, Faculty of Economics and Administrative Sciences, Karadeniz Technical University , Trabzon , Turkey
Wan Shibiao
Electronic publication date: 2025 Jun 18
Publication date: 2025
Volume: 11
Electronic Location ID: e2970
Received 2025 Feb 24; Accepted 2025 May 28
Copyright: © 2025 Gurcan
Copyright year: 2025
Copyright holder: Gurcan
License: This is an open access article distributed under the terms of the Creative Commons Attribution License, which permits unrestricted use, distribution, reproduction and adaptation in any medium and for any purpose provided that it is properly attributed. For attribution, the original author(s), title, publication source (PeerJ Computer Science) and either DOI or URL of the article must be cited.
License URL: https://creativecommons.org/licenses/by/4.0/

Keywords: B-cell epitope prediction, Ensemble learning, Resampling techniques, Immunoinformatics, Vaccine design

Funding: The author has received no funding for this work.

==============================
This study presents a comprehensive machine learning framework to enhance the prediction accuracy of B-cell epitopes and antibody recognition related to Severe Acute Respiratory Syndrome (SARS) and Coronavirus Disease 2019 (COVID-19). To address the issue of data imbalance, various resampling techniques were applied using three types of strategies: over-sampling, under-sampling, and hybrid-sampling. The implemented resampling methods were designed to improve class balance and enhance model training. The rebalanced datasets were then used in model building with ensemble classifiers employing Boosting, Bagging, and Balancing strategies. Hyperparameter optimization for the classifiers was conducted using GridSearchCV, while feature selection was performed with the recursive feature elimination (RFE) algorithm. Model performance was evaluated using seven different metrics: Accuracy, Precision, Recall, F1-score, receiver operating characteristic area under the curve (ROC AUC), precision recall area under the curve (PR AUC), and Matthews correlation coefficient (MCC). Furthermore, statistical significance analyses including paired t-test, Wilcoxon, and permutation tests confirmed the reliability of the model improvements. To interpret the model’s predictive behavior, peptides with the highest confidence among correctly classified instances were identified as potential epitope candidates. The results indicate that the combination of Synthetic Minority Over-Sampling Technique—Edited Nearest Neighbors (SMOTE-ENN), and ExtraTrees yielded the best performance, achieving an ROC AUC score of 0.9899. The combination of Instance Hardness Threshold (IHT) and ExtraTrees followed closely with a score of 0.9799. These findings emphasize the effectiveness of integrating resampling models and balancing ensemble classifiers in improving the accuracy of B-cell epitope prediction and antibody recognition for SARS and COVID-19 infections. This study contributes to vaccine development efforts and the advancement of immunoinformatics research by identifying promising epitope candidates.

Introduction

The COVID-19 pandemic has challenged global public health, overwhelming healthcare systems and highlighting the urgent need for rapid vaccine development to control the virus and protect populations. Effective vaccines have become a cornerstone in the fight against the pandemic. The immune system’s ability to recognize viruses and initiate defensive mechanisms plays a critical role in vaccine development (Cunningham et al., 2016; Pennisi, Genovese & Gianfredi, 2024). In particular, B-cells are essential components of the immune system, capable of recognizing antigens and triggering immune responses (Sanchez-Trincado, Gomez-Perosanz & Reche, 2017; Bukhari et al., 2022; Pennisi, Genovese & Gianfredi, 2024). B-cells identify specific regions of antigenic proteins, known as epitopes, and produce large quantities of antibodies (Jespersen et al., 2019; Bukhari et al., 2022). As part of the immune defense, these antibodies play a crucial role in combating infections. Therefore, accurately identifying antigenic epitopes is a key determinant in the design of effective vaccines (El-Manzalawy, Dobbs & Honavar, 2017; Sanchez-Trincado, Gomez-Perosanz & Reche, 2017; Thomas et al., 2022). Epitopes not only elicit immune responses but also activate defense mechanisms that prevent the further spread of viruses in the body (Maloney et al., 2020). Antibodies bind to these epitopes, neutralizing viruses and inhibiting their ability to attach to host cells. As such, the accurate identification of epitopes directly impacts vaccine efficacy (Correia et al., 2014; Palma, 2023). The proper selection of antigens and the identification of epitopes that most effectively trigger immune responses are foundational to the rapid and effective vaccine development process (Cunningham et al., 2016; El-Manzalawy, Dobbs & Honavar, 2017; Sanchez-Trincado, Gomez-Perosanz & Reche, 2017; Palma, 2023).

Viruses such as SARS-CoV and SARS-CoV-2, which cause SARS and COVID-19, respectively, possess specific antigenic regions (epitopes) recognizable by B-cells, which are of paramount importance (Amrun et al., 2020; Wang et al., 2020; Bukhari et al., 2022; Palma, 2023). Accurate prediction of these antigenic regions is critical for designing effective vaccines and enhancing immune responses (El-Manzalawy, Dobbs & Honavar, 2017; Jespersen et al., 2019; Wang et al., 2020). B-cell epitopes, defined as specific regions of antigenic proteins recognized by B-cells and bound by antibodies, play a pivotal role in neutralizing antigens and bolstering the immune response (Jespersen et al., 2019; Lim et al., 2022). Mimicking the structure and function of epitopes within a vaccine aims to induce specific antibody production in the host organism (Xu & Kulp, 2019; Lim et al., 2022). However, predicting epitopes accurately is often a complex and time-consuming process (Sanchez-Trincado, Gomez-Perosanz & Reche, 2017; Amrun et al., 2020; Wang et al., 2020; Bukhari et al., 2022). Traditional methods, including X-ray crystallography, nuclear magnetic resonance (NMR) spectroscopy, and cryo-electron microscopy (cryo-EM), are widely employed to analyze the three-dimensional structures of antigen-antibody interactions (Toride King & Brooks, 2018; Xu & Kulp, 2019; Grewal, Hegde & Yanow, 2024). In addition, several previous B-cell epitope prediction studies for SARS-CoV-2 have predominantly employed sequence-based strategies, including alignment, mutation frequency analysis, and motif discovery, relying heavily on classical bioinformatics techniques (Sanchez-Trincado, Gomez-Perosanz & Reche, 2017; Agarwal et al., 2022; Tai, Li & Zhang, 2023). In some cases, efforts have been limited to the construction of epitope databases based solely on computational predictions without incorporating advanced machine learning approaches (Chen et al., 2020; Kiyotani et al., 2020). Representative examples of these traditional approaches include large-scale genome alignment to detect conserved epitopes (Agarwal et al., 2022), computational annotation and database creation for spike-derived epitopes (Tai, Li & Zhang, 2023), structural modeling to propose new B and T cell epitopes from critical protein regions (Dawood et al., 2021), and high-resolution epitope motif mapping in clinical cohorts using proteome-wide screening tools (Haynes et al., 2021).

While these traditional techniques provide highly accurate structural data, they are costly and time-intensive, limiting their widespread use. In contrast, recent computational approaches like AlphaFold (AF2/AF3) offer rapid, cost-effective predictions of antigen-antibody interactions, though experimental validation remains essential (Sanchez-Trincado, Gomez-Perosanz & Reche, 2017; Jespersen et al., 2019; Bukhari et al., 2022; Thomas et al., 2022). This has driven the search for alternative methods that are both faster and more cost-effective, particularly in the context of urgent vaccine development. In recent years, bioinformatics and machine learning approaches have emerged as powerful tools to address these challenges (El-Manzalawy, Dobbs & Honavar, 2017; Sanchez-Trincado, Gomez-Perosanz & Reche, 2017; Wang et al., 2020; Bukhari et al., 2022; Thomas et al., 2022; Gurcan, 2023; Kim et al., 2024). Computational B-cell epitope prediction methods offer a faster, more economical, and more efficient alternative to traditional experimental techniques (Sanchez-Trincado, Gomez-Perosanz & Reche, 2017; Jespersen et al., 2019; Thomas et al., 2022; Zheng, Liang & Zhang, 2023; Grewal, Hegde & Yanow, 2024; Kim et al., 2024; Pennisi, Genovese & Gianfredi, 2024).

Considering this background, the present study introduces a comprehensive methodology integrating resampling techniques (Instance Hardness Threshold (IHT), repetitively enhanced neural network (RENN), Synthetic Minority Over-Sampling Technique—Edited Nearest Neighbors (SMOTE-ENN), etc.) with ensemble learning models to improve the prediction accuracy of B-cell epitopes, focusing on SARS-CoV and SARS-CoV-2, the causative agents of SARS and COVID-19. The study uses robust datasets obtained from reliable sources such as the Immune Epitope Database (IEDB) and UniProt, encompassing protein and peptide characteristics that influence epitope-antibody interactions. The proposed methodology incorporates not only target amino acid sequences but also structural and chemical properties of antigenic proteins, enabling a more holistic analysis. The findings demonstrate that the proposed methodology contributes significantly to biomedical research and vaccine development, particularly in scenarios demanding rapid responses to infectious diseases. By mitigating data imbalances and enhancing model generalizability, this study advances epitope prediction accuracy for the design of COVID-19 and SARS vaccines, supporting data-driven vaccine development. In summary, the main contributions of this study are as follows: This study introduces a machine learning framework combining resampling techniques with ensemble classifiers for B-cell epitope prediction in SARS-CoV and SARS-CoV-2.

New features were derived from protein-peptide and epitope-antibody interaction data, incorporating both sequence-based and physicochemical attributes.

The approach enables broader analysis by integrating structural and biochemical characteristics alongside amino acid sequences.

To address class imbalance, a range of resampling techniques (over-sampling, under-sampling, and hybrid-sampling) was applied, thereby enhancing model performance and robustness.

Three types of ensemble classifiers—Boosting, Bagging, and Balancing—were employed to enhance predictive performance and enable inclusive comparison across modeling strategies.

Hyperparameter optimization was performed using GridSearchCV for all ensemble classifiers to determine the best parameter settings and ensure fair comparison across models.

Recursive feature elimination (RFE) was applied to select the most informative features, thereby improving model interpretability.

Model performance was comprehensively evaluated using seven different metrics: Accuracy, Precision, Recall, F1-score, receiver operating characteristic area under the curve (ROC AUC), precision recall area under the curve (PR AUC), and Matthews correlation coefficient (MCC).

Model comparisons were supported by statistical significance tests such as paired t-test, Wilcoxon signed-rank test, and permutation test, confirming that the observed improvements were statistically meaningful rather than random.

Confidence-based analysis identified high-confidence correct predictions as candidate epitopes, which require further biological validation.

The structure of the article is as follows: “Materials and Methods” describes the materials and methods, including a detailed explanation of the dataset, data preprocessing steps, feature extraction techniques, and resampling methods, as well as an overview of the ensemble model architecture, which incorporates bagging, boosting, and balancing strategies. “Experimental Results” presents the performance evaluation, showcasing the results from individual models, resampling methods, and ensemble approaches, followed by a comparative analysis of their effectiveness. “Discussion” discusses the findings, offering an interpretation of the results, validating the outcomes, identifying the study’s limitations, and exploring the broader implications for epitope prediction. Finally, “Conclusion” concludes the article and suggests potential avenues for future research in the prediction of SARS-CoV and SARS-CoV-2 epitopes using advanced machine learning approaches.

Materials and Methods

This section outlines the methodology employed in this study, encompassing several critical components: dataset overview, feature engineering and data preprocessing, implementation of resampling techniques, deployment of ensemble classifiers, experimental configuration, and performance assessment. To provide a clear and structured presentation of the process, the methodology is organized into the following subheadings, detailing each step in the workflow.

Dataset description

The empirical dataset used in this study is the B-cell epitope dataset, developed during research into epitope prediction and vaccine design (Noumi et al., 2021). The dataset is publicly available on Kaggle, thereby promoting reproducibility and transparency in scientific research (Noumi et al., 2021; Kaggle, 2024). This dataset provides detailed information about peptides and their corresponding proteins, alongside physicochemical features relevant for epitope prediction (Jain et al., 2021; Noumi et al., 2021). The dataset is derived from IEDB and UniProt. It contains 14,387 rows representing all combinations of 14,362 unique peptides and 757 proteins (Kaggle, 2024). The dataset includes 13 independent variables (features) and 1 dependent variable (target). The dataset includes peptide-level features such as the start and end positions that define the peptide’s location within the parent protein, and the amino acid sequence of the peptide. It also includes physicochemical properties like chou_fasman (β-turn propensity), emini (relative surface accessibility), kolaskar_tongaonkar (antigenicity), and parker (hydrophobicity). In terms of protein-level features, the dataset provides the parent protein’s identifier and its amino acid sequence, as well as properties such as isoelectric point, aromaticity, hydrophobicity, and stability (El-Manzalawy, Dobbs & Honavar, 2017; Noumi et al., 2021; Ozger & Cihan, 2022). For this experimental dataset, the target variable indicates the binary classification of antibody valence, distinguishing between positive and negative classes. Specifically, the dataset contains 10,485 rows labeled as negative (0) and 3,902 rows labeled as positive (1), highlighting a significant class imbalance (Kaggle, 2024). This imbalance can negatively impact the performance of machine learning models by biasing predictions toward the majority class (Khushi et al., 2021; Kraiem, Sánchez-Hernández & Moreno-García, 2021; Wang & Cheng, 2021; Gurcan & Soylu, 2024a). To address this issue, resampling techniques were employed to balance the dataset, ensuring that the model can learn effectively from both classes and improve its generalization capability (Gurcan & Soylu, 2024b).

Feature engineering and data preprocessing

Robust data preprocessing (cleaning, normalization) and feature engineering are essential for optimizing dataset usability and predictive accuracy in machine learning (Haixiang et al., 2017; Plotnikova, Dumas & Milani, 2020; Ahsan et al., 2021). To enhance the predictive capability of the model, several additional features were derived from the original dataset. The protein sequence length was calculated as the length of the protein_seq column, while the peptide sequence length was determined as the length of the peptide_seq column. The parent protein ID length was obtained as the length of the parent_protein_id column. Additionally, the peptide length was computed as the difference between the end_position and start_position columns, plus one (Caoili, 2022). To ensure that categorical variables could be effectively used in machine learning models, all string-based categorical features were converted into numerical format using label encoding. Specifically, the columns “parent_protein_id”, “peptide_seq”, and “protein_seq” were encoded as integers to enable compatibility with the classification models (Raschka & Mirjalili, 2019; Raschka, Patterson & Nolet, 2020; Gurcan, 2023; Gurcan & Soylu, 2024b). These engineered features captured structural and positional variations in the peptide and protein sequences, enriching the feature space for model training (Jespersen et al., 2019; Amrun et al., 2020; Bukhari et al., 2022). As a result, the total number of features increased from 14 to 18, providing the model with a more comprehensive representation of the data (Jespersen et al., 2019; Ozger & Cihan, 2022; Gurcan, 2023).

After new features were incorporated into the dataset, recursive feature elimination (RFE) was employed to identify the most relevant features for B-cell epitope prediction. RFE is a feature selection method that iteratively removes less important features to select those that contribute most significantly to the model’s predictive performance. The RFE algorithm was implemented using the Random Forest classifier as the base estimator. Feature selection was performed on the original (baseline) dataset without applying any resampling techniques. The optimal number of features was determined based on the ROC AUC scores calculated for various feature subsets. At the end of the RFE process, all variables were ranked by their importance, and the subset of features that yielded the highest ROC AUC score was selected as the optimal feature set.

The relationship between the ROC AUC score and the number of selected features is visualized in Fig. 1. As shown in the figure, the highest performance was achieved using nine features. The top nine selected features identified by RFE are as follows: “parent_protein_id”, “peptide_seq_length”, “hydrophobicity”, “isoelectric_point”, “emini”, “parker”, “end_position”, “start_position”, and “protein_seq_length”. The eight features eliminated by the RFE algorithm were: “chou_fasman”, “aromaticity”, “protein_seq”, “kolaskar_tongaonkar”, “stability”, “peptide_seq”, “peptide_length”, and “parent_protein_id_length”. This process identified the most informative features and optimized their number, resulting in a refined feature matrix X for robust model performance.

Figure 1 Relationship between number of selected features and ROC AUC score using RFE.

The target column, target, was retained as the response variable. To standardize the feature space and accommodate varying scales among variables, Min-Max Scaling was applied, ensuring that all features are transformed into a common range between 0 and 1 (Haixiang et al., 2017; Ahsan et al., 2021; Gurcan, 2024; Scikit-learn, 2024). Additionally, to address the class imbalance in the target variable, several resampling techniques were applied, including oversampling the minority class and undersampling the majority class (Kraiem, Sánchez-Hernández & Moreno-García, 2021; Wang, Wang & He, 2023; Gurcan & Soylu, 2024a). Furthermore, hybrid-sampling approaches were utilized, combining both oversampling and undersampling methods (Khushi et al., 2021; Palli et al., 2022). The dataset was further divided into training and validation sets to ensure robust evaluation of the model. The train_test_split function was utilized to split the resampled data into training and testing subsets, with 80% of the data used for training and 20% reserved for testing. Stratified sampling (stratify=y_resampled) was employed during the split to preserve the class distribution in the target variable across both subsets, ensuring a balanced representation of classes (Pedregosa et al., 2011; Raschka & Mirjalili, 2019; Scikit-learn, 2024).

Application of resampling techniques

Resampling techniques are crucial in machine learning for addressing class imbalance, ensuring models perform well on minority classes (Wang et al., 2020; Gurcan & Soylu, 2024b). This is particularly important in B-cell epitope prediction, where balanced data improves classifier accuracy in identifying antibody-inducing regions (Ozger & Cihan, 2022). In this study, we implement eight resampling methods to address class imbalance, categorized into three types: over-sampling, under-sampling, and hybrid-sampling, along with a Baseline (no resampling) scenario for comparison (Khushi et al., 2021; Palli et al., 2022; Gurcan & Soylu, 2024a). Over-sampling methods aim to increase the representation of the minority class and include KM-SMOTE, which uses k-means clustering to generate synthetic samples more effectively (Xu et al., 2021; Gurcan & Soylu, 2024a), Synthetic Minority Oversampling Technique (SMOTE), which creates new samples by interpolating between existing ones (Zheng, Cai & Li, 2015; Kraiem, Sánchez-Hernández & Moreno-García, 2021), and Random Oversampling (ROS), which simply duplicates samples from the minority class (Khushi et al., 2021; Kraiem, Sánchez-Hernández & Moreno-García, 2021; Gurcan & Soylu, 2024a).

Under-sampling methods, designed to reduce the majority class representation, include IHT, which removes samples based on their classification difficulty (Wang, Wang & He, 2023; Gurcan & Soylu, 2024a), RENN, which eliminates majority samples misclassified by their nearest neighbors (Kim & Hwang, 2022; Gurcan & Soylu, 2024a), and ENN (edited nearest neighbor), which removes conflicting majority samples to improve class separation (Kraiem, Sánchez-Hernández & Moreno-García, 2021; Gurcan & Soylu, 2024a). Hybrid-sampling approaches integrate the strengths of both over-sampling and under-sampling methods to achieve better balance and improve model performance. Hybrid-sampling combines both strategies for greater balance, with SMOTE-ENN using SMOTE for generating synthetic samples and ENN for refining the majority class (Fotouhi, Asadi & Kattan, 2019; Kraiem, Sánchez-Hernández & Moreno-García, 2021), and SMOTE-TL, which integrates SMOTE with Tomek Link removal to enhance class boundaries (Khushi et al., 2021; Swana, Doorsamy & Bokoro, 2022). The Baseline scenario, which does not apply any resampling, serves as a reference point to assess the impact of these methods on model performance (Gurcan & Soylu, 2024a, 2024b). In summary, Table 1 provides an overview of how each resampling technique modifies the class distribution and addresses the imbalance ratio, highlighting their adjustments to the dataset.

Table 1 Class distributions and imbalance adjustments by resampling models.

Sampling type	Resampling method	Class count (Majority)	Class count (Minority)	Imbalance ratio	
No resampling	Baseline	10,485	3,902	2.69	
Over-sampling	SMOTE	10,485	10,485	1.00	
KM-SMOTE	10,485	10,488	1.00	
ROS	10,485	10,485	1.00	
Under-sampling	ENN	8,188	3,902	2.10	
RENN	7,473	3,902	1.92	
IHT	4,525	3,902	1.16	
Hybrid-sampling	SMOTE-ENN	7,931	8,481	0.94	
SMOTE-TL	10,251	10,251	1.00	

The Baseline model, which does not apply any resampling, maintains the original class imbalance with a ratio of 2.69. Over-sampling techniques, such as SMOTE, K-Means Synthetic Minority Oversampling Technique (KM-SMOTE), and ROS, effectively balance the classes by increasing the minority class count to match the majority, achieving a ratio of 1.0. Under-sampling methods, including ENN, RENN, and IHT, reduce the majority class size, with IHT achieving the lowest imbalance ratio of 1.16. Hybrid approaches, such as SMOTE-ENN and SMOTE-TL, combine over- and under-sampling to adjust class distributions, with SMOTE-ENN slightly favoring the minority class and achieving a ratio of 0.94, while SMOTE-TL fully balances the dataset with a ratio of 1.0. These resampling strategies highlight their diverse impacts on class balance, shaping the dataset’s distribution and enhancing its suitability for effective model training (Kraiem, Sánchez-Hernández & Moreno-García, 2021; Wang, Wang & He, 2023; Gurcan & Soylu, 2024a).

Application of ensemble classifiers with hyperparameter optimization

In this study on B-cell epitope prediction, we employed ten ensemble models across three categories: Bagging, Boosting, and Balancing (Wang & Cheng, 2021; Gurcan & Soylu, 2024b, 2024a). Each category represents a distinct ensemble learning approach, designed to enhance model performance and address challenges such as class imbalance. Bagging, or Bootstrap Aggregating, reduces variance by training multiple models on random subsets of the data and aggregating their predictions (Gurcan & Soylu, 2024b, 2024a). Random Forest builds multiple decision trees and combines their outputs through majority voting, offering robustness against overfitting. ExtraTrees is similar to Random Forest but introduces randomness in feature splits to improve computational efficiency (Ahsan et al., 2021; Gurcan, 2024). The Bagging Classifier serves as a general-purpose ensemble method that combines predictions from base models trained on bootstrap samples (Plaia et al., 2022). Boosting focuses on improving model accuracy by sequentially training weak learners, with each iteration correcting the errors of the previous model (Al-Azzam & Shatnawi, 2021; Wang, Wang & He, 2023; Gurcan & Soylu, 2024b). Extreme gradient boosting (XGBoost) is an efficient gradient boosting framework optimized for performance on structured data. CatBoost specializes in handling categorical features with minimal preprocessing while maintaining high accuracy. Light Gradient-Boosting Machine (LightGBM) uses a leaf-wise tree growth strategy, enabling faster training and better scalability for large datasets (Gurcan & Soylu, 2024a, 2024b).

Balancing integrates resampling strategies within ensemble methods to address class imbalance, ensuring fair treatment of both classes (Khushi et al., 2021; Gurcan & Soylu, 2024a, 2024b). Balanced Random Forest (BalancedRF) incorporates under-sampling of the majority class during tree construction to improve minority class representation (Su et al., 2015). EasyEnsemble generates multiple balanced subsets of the majority class, training a separate model on each to mitigate bias (Wang & Cheng, 2021). Balanced Bagging (BalancedBG) combines under-sampling with bagging techniques to create diverse and unbiased ensembles (Khushi et al., 2021; Gurcan & Soylu, 2024a). Finally, Random under-sampling integrated in the learning of AdaBoost (RUSBoost) blends under-sampling with boosting, focusing on improving predictions for the minority class (Gurcan & Soylu, 2024a).

In an effort to enhance the performance of B-cell epitope prediction models, hyperparameter optimization for the ensemble classifiers used in this study was performed using the GridSearchCV technique. This method is a widely used, systematic, and exhaustive search method for tuning hyperparameters in machine learning models. It evaluates all possible combinations within predefined parameter ranges to identify the optimal set of hyperparameters for each model. During the optimization process, the ROC AUC score was used as the evaluation metric. This approach helps to reduce the risk of overfitting while enhancing the generalization ability of the models. The parameter grids for each classifier were determined based on prior literature and preliminary analyses, and a five-fold cross-validation (5-fold CV) was applied to ensure the reliability of the results. The best parameters obtained via GridSearchCV are presented in Table 2 and were used in the final model evaluations. Combining these approaches with fine-tuned hyperparameters enables the development of robust and balanced ensemble-based predictions for B-cell epitope identification.

Table 2 Best-performing hyperparameters for ensemble models from GridSearchCV optimization.

Type	Model	Best parameters	
Boosting	XGBoost	n_estimators=100, learning_rate=0.3, max_depth=6, booster=‘gbtree’, subsample=1.0, colsample_bytree=1.0, tree_method=‘auto’, gamma=0, reg_alpha=0, reg_lambda=1, random_state=42	
CatBoost	iterations=1000, learning_rate=0.03, depth=6, loss_function=‘Logloss’, l2_leaf_reg=3, border_count=254, grow_policy=‘SymmetricTree’, thread_count=−1, verbose=0, random_state=42	
LightGBM	learning_rate=0.1, n_estimators=200, num_leaves=50, boosting_type=‘gbdt’, max_depth=−1, objective=‘binary’, subsample=1.0, colsample_bytree=1.0, reg_alpha=0.0, reg_lambda=0.0, random_state=42	
Bagging	Random Forest	n_estimators=200, max_depth=None, max_features=‘sqrt’, min_samples_split=2, criterion=‘gini’, min_samples_leaf=1, bootstrap=True, min_weight_fraction_leaf=0.0, max_leaf_nodes=None, random_state=42	
ExtraTrees	n_estimators=200, max_features=‘sqrt’, criterion=‘gini’, max_depth=None, bootstrap=False, min_samples_split=2, min_samples_leaf=1, max_leaf_nodes=None, min_impurity_decrease=0.0, random_state=42	
Bagging	n_estimators=20, bootstrap=True, bootstrap_features=False, max_samples=1.0, max_features=1.0, n_jobs=None, oob_score=False, warm_start=False, verbose=0, random_state=42	
Balancing	BalancedRF	sampling_strategy=‘all’, replacement=True, bootstrap=False, n_estimators=100, max_features=‘sqrt’, criterion=‘gini’, min_samples_split=2, min_samples_leaf=1, max_depth=None, max_leaf_nodes=None, random_state=42	
EasyEnsemble	sampling_strategy=‘all’, n_estimators=10, replacement=False, warm_start=False, n_jobs=None, verbose=0, random_state=42	
BalancedBG	n_estimators=20, sampling_strategy=‘all’, replacement=True, bootstrap=False, warm_start=False, n_jobs=None, verbose=0, random_state=42	
RUSBoost	n_estimators=100, sampling_strategy=‘all’, learning_rate=1.0, algorithm=‘SAMME.R’, random_state=42	

Experimental setup and performance evaluation

The experimental setup for this study was conducted in a high-performance computing environment to ensure efficiency and stability. The system featured an Intel i7-12650H processor, Nvidia RTX 3060 GPU, and 16 GB of RAM, operating on a 64-bit version of Windows 10. Python 3.12, along with Jupyter Notebook, was utilized for the development and execution of the analysis (Pedregosa et al., 2011; Nelli, 2023). The selected tools and libraries ensured smooth execution throughout the machine learning pipeline, covering data preprocessing, model training, evaluation, and deployment. Python’s versatility and extensive ecosystem made it the perfect choice for this research, enabling efficient handling of complex tasks in machine learning and data science (Raschka, Patterson & Nolet, 2020; Gurcan, 2023; Nelli, 2023; Gurcan & Soylu, 2024a).

In this study, the performance evaluation was based on seven key metrics: Accuracy, Precision, Recall, F1-score, PR AUC, MCC, and ROC AUC (Gurcan & Soylu, 2024b; Disci, Gurcan & Soylu, 2025). Using these seven metrics allows for a comprehensive evaluation, ensuring that various aspects of model performance are considered, particularly in the presence of class imbalance (Nelli, 2023; Gurcan & Soylu, 2024a; Scikit-learn, 2024). Accuracy measures the overall proportion of correct predictions made by the model. Precision indicates the proportion of true positive predictions among all positive predictions (Kim & Hwang, 2022; Gurcan & Soylu, 2024b). Recall represents the proportion of true positives identified out of all actual positive instances (Kim & Hwang, 2022; Kaya & Gürsoy, 2023). F1-score provides a balanced measure between precision and recall, especially in cases of imbalanced classes (Al-Azzam & Shatnawi, 2021; Kaya & Gürsoy, 2023). PR AUC measures model performance under class imbalance by evaluating precision-recall trade-offs across thresholds. MCC evaluates classification quality by considering all confusion matrix categories, making it robust for imbalanced datasets. Finally, ROC AUC evaluates the model’s ability to discriminate between positive and negative classes, with higher values indicating better classification performance (Kraiem, Sánchez-Hernández & Moreno-García, 2021; Gurcan & Soylu, 2024b). This metric evaluates how well the classifier can separate the two classes across various threshold levels, making it a reliable indicator of model performance, even when the class distribution is uneven (Walsh & Tardy, 2023; Gurcan & Soylu, 2024b). Focusing on ROC AUC ensures that the models are not biased toward the majority class and can effectively identify rare events, such as the presence of antibodies in B-cell epitope prediction (Noumi et al., 2021; Walsh & Tardy, 2023; Gurcan & Soylu, 2024a). Finally, to assess whether the observed performance improvements between models resulted from true model enhancements rather than random variation or changes in data structure, three statistical significance tests were applied to the obtained results. Based on the ROC AUC scores derived from five-fold cross-validation, three widely accepted statistical tests, including the paired t-test, the Wilcoxon signed-rank test, and the permutation test, were conducted. Furthermore, confidence-based analyses were conducted to identify peptides most confidently and correctly predicted as potential epitope candidates, while high-confidence misclassifications were examined solely to assess model limitations. In the results section, all classifier performance tables, including those for resampling methods, are organized and presented in descending order based on ROC AUC scores to provide a clear and consistent comparison of model performance.

Experimental results

Evaluation of classifiers without resampling strategies

In this analysis, classifiers are evaluated in the baseline scenario without any resampling methods applied, focusing on various performance metrics. The metrics include: Accuracy, Precision, Recall, F1-score, PR AUC, MCC, and ROC AUC. Table 3 presents the performance metrics of classifiers in the baseline scenario, organized from highest to lowest ROC AUC scores, showcasing each model’s ability to discriminate between positive and negative classes in the B-cell epitope prediction task. When evaluating ROC AUC performance, BalancedRF (0.8640) emerges as the top-performing classifier, followed closely by ExtraTrees (0.8572) and BalancedBG (0.8560). Random Forest (0.8529) also demonstrates strong discriminative capability, while bagging-based methods (e.g., Bagging: 0.8312) consistently outperform boosting approaches like LightGBM (0.8199), XGBoost (0.8144), and CatBoost (0.7896). Hybrid balancing techniques such as RUSBoost (0.7445) and EasyEnsemble (0.7131) lag behind, highlighting the superiority of bagging with intrinsic class balancing (e.g., BalancedRF) for imbalanced classification tasks.

Table 3 Performance metrics of classifiers in the baseline scenario without resampling.

Classifier	Accuracy	Precision	Recall	F1-score	PR AUC	MCC	ROC AUC	
BalancedRF	0.8697	0.7197	0.8515	0.7801	0.6531	0.6933	0.8640	
ExtraTrees	0.8867	0.7905	0.7926	0.7916	0.6829	0.7138	0.8572	
BalancedBG	0.8662	0.7185	0.8335	0.7718	0.6441	0.6816	0.8560	
Random Forest	0.8864	0.7971	0.7798	0.7883	0.6813	0.7108	0.8529	
Bagging	0.8735	0.7829	0.7388	0.7602	0.6493	0.6749	0.8312	
LightGBM	0.8711	0.7945	0.7081	0.7488	0.6418	0.6645	0.8199	
XGBoost	0.8659	0.7817	0.7017	0.7395	0.6295	0.6513	0.8144	
CatBoost	0.8544	0.7785	0.6479	0.7072	0.5999	0.6160	0.7896	
RUSBoost	0.7589	0.5424	0.7132	0.6162	0.4646	0.4539	0.7445	
EasyEnsemble	0.7248	0.4949	0.6876	0.5756	0.4251	0.3911	0.7131	

Evaluation of classifiers for over-sampling strategies

This analysis evaluates the performance of ten classifiers applied with three over-sampling methods (ROS, SMOTE, and KM-SMOTE) for B-cell epitope classification, focusing on their ability to address class imbalance effectively. Table 4 presents the top five classifiers for each oversampling method, ranked by multiple performance metrics and systematically ordered by descending ROC AUC values. In the ROS model, classifiers achieved the highest overall ROC AUC scores, with ExtraTrees (0.9466) and BalancedRF (0.9404) demonstrating the best performance. The KM-SMOTE model showed slightly lower ROC AUC values compared to ROS, where Extra Trees (0.9240) and Random Forest (0.9213) were the top performers. SMOTE provided a balanced but comparatively lower performance, with ExtraTrees (0.9163) and Random Forest (0.9139) achieving the highest ROC AUC scores in this category. As a result, the ROS model, particularly when paired with ExtraTrees, BalancedRF, and Random Forest classifiers, proved to be the most effective resampling approach for improving class discrimination.

Table 4 Performance metrics of classifiers using over-sampling methods.

Sampling	Classifier	Accuracy	Precision	Recall	F1-score	PR AUC	MCC	ROC AUC	
ROS	ExtraTrees	0.9466	0.9171	0.9819	0.9484	0.9096	0.8954	0.9466	
BalancedRF	0.9404	0.9066	0.9819	0.9428	0.8993	0.8838	0.9404	
Random Forest	0.9385	0.9045	0.9804	0.9410	0.8966	0.8801	0.9385	
Bagging	0.9351	0.9029	0.9752	0.9376	0.8929	0.8731	0.9351	
BalancedBG	0.9351	0.9029	0.9752	0.9376	0.8929	0.8731	0.9351	
KM-SMOTE	ExtraTrees	0.9240	0.9222	0.9261	0.9241	0.8910	0.8479	0.9240	
Random Forest	0.9213	0.9234	0.9190	0.9212	0.8891	0.8427	0.9213	
BalancedRF	0.9199	0.9203	0.9194	0.9199	0.8865	0.8398	0.9199	
Bagging	0.9139	0.9243	0.9018	0.9129	0.8826	0.8281	0.9139	
BalancedBG	0.9123	0.9215	0.9013	0.9113	0.8800	0.8248	0.9123	
SMOTE	ExtraTrees	0.9163	0.8986	0.9385	0.9181	0.8741	0.8334	0.9163	
Random Forest	0.9139	0.8953	0.9375	0.9159	0.8706	0.8288	0.9139	
BalancedRF	0.9139	0.8963	0.9361	0.9158	0.8710	0.8287	0.9139	
Bagging	0.9099	0.8988	0.9237	0.9111	0.8684	0.8201	0.9099	
BalancedBG	0.9096	0.8984	0.9237	0.9109	0.8680	0.8196	0.9096	

Evaluation of classifiers for under-sampling strategies

This analysis examines the performance of ten classifiers applied with three under-sampling methods (IHT, RENN, and ENN) for B-cell epitope classification, focusing on their effectiveness in managing class imbalance. Table 5 displays the top five classifiers for each under-sampling technique, ranked by multiple performance metrics and ordered by descending ROC AUC values to highlight their classification performance. In the IHT model, classifiers achieved the highest overall ROC AUC scores among the under-sampling methods, with ExtraTrees (0.9799) and Random Forest (0.9787) leading the performance. The RENN model demonstrated slightly lower ROC AUC values compared to IHT, with ExtraTrees (0.9695) and BalancedRF (0.9595) as the top performers. The ENN model showed a balanced yet relatively lower performance, where ExtraTrees (0.9509) and BalancedRF (0.9486) ranked highest in ROC AUC. In summary, the IHT model, particularly in combination with ExtraTrees, Random Forest, and BalancedRF classifiers, emerged as the most effective under-sampling approach to improve classification performance.

Table 5 Performance metrics of classifiers using under-sampling methods.

Sampling	Classifier	Accuracy	Precision	Recall	F1-score	PR AUC	MCC	ROC AUC	
IHT	ExtraTrees	0.9801	0.9857	0.9731	0.9794	0.9722	0.9602	0.9799	
Random Forest	0.9789	0.9819	0.9744	0.9781	0.9692	0.9577	0.9787	
BalancedRF	0.9770	0.9794	0.9731	0.9762	0.9660	0.9540	0.9769	
LightGBM	0.9764	0.9806	0.9705	0.9755	0.9660	0.9528	0.9762	
BalancedBG	0.9758	0.9805	0.9692	0.9749	0.9653	0.9515	0.9756	
RENN	ExtraTrees	0.9759	0.9776	0.9501	0.9636	0.9456	0.9459	0.9695	
BalancedRF	0.9660	0.9582	0.9398	0.9489	0.9208	0.9236	0.9595	
Random Forest	0.9686	0.9771	0.9283	0.9521	0.9311	0.9294	0.9586	
Bagging	0.9609	0.9650	0.9168	0.9402	0.9126	0.9118	0.9500	
BalancedBG	0.9561	0.9473	0.9206	0.9338	0.8988	0.9012	0.9474	
ENN	ExtraTrees	0.9600	0.9476	0.9257	0.9365	0.9009	0.9074	0.9509	
BalancedRF	0.9550	0.9285	0.9309	0.9297	0.8864	0.8966	0.9486	
Random Forest	0.9546	0.9479	0.9078	0.9274	0.8899	0.8949	0.9422	
BalancedBG	0.9481	0.9236	0.9129	0.9182	0.8710	0.8802	0.9388	
Bagging	0.9489	0.9432	0.8937	0.9178	0.8769	0.8815	0.9343	

Evaluation of classifiers for hybrid-sampling strategies

This analysis evaluates the performance of ten classifiers applied with two hybrid-sampling methods (SMOTE-ENN and SMOTE-TL) for B-cell epitope classification, emphasizing their ability to manage class imbalance. Table 6 presents the top five classifiers for each hybrid-sampling model, ranked based on multiple performance metrics and organized in descending order of ROC AUC. In the SMOTE-ENN model, classifiers achieved the highest overall ROC AUC scores among the hybrid-sampling methods, with Extra Trees (0.9899) and Random Forest (0.9856) demonstrating superior performance. The SMOTE-TL model showed slightly lower ROC AUC values compared to SMOTE-ENN, where ExtraTrees (0.9363) and BalancedRF (0.9318) were the top performers. Experimental results identified SMOTE-ENN as the most effective hybrid-sampling technique, particularly when integrated with ExtraTrees, Random Forest, and BalancedRF classifiers.

Table 6 Performance metrics of classifiers using hybrid-sampling models.

Sampling	Classifier	Accuracy	Precision	Recall	F1-score	PR AUC	MCC	ROC AUC	
SMOTE-ENN	ExtraTrees	0.9899	0.9885	0.9915	0.9900	0.9844	0.9798	0.9899	
Random Forest	0.9856	0.9831	0.9885	0.9858	0.9776	0.9712	0.9856	
BalancedRF	0.9853	0.9831	0.9879	0.9855	0.9773	0.9706	0.9853	
Bagging	0.9816	0.9824	0.9812	0.9818	0.9734	0.9632	0.9816	
BalancedBG	0.9813	0.9830	0.9800	0.9815	0.9734	0.9626	0.9813	
SMOTE-TL	ExtraTrees	0.9362	0.9161	0.9605	0.9377	0.8996	0.8735	0.9363	
BalancedRF	0.9318	0.9118	0.9560	0.9334	0.8937	0.8646	0.9318	
Random Forest	0.9318	0.9126	0.9550	0.9333	0.8940	0.8645	0.9318	
BalancedBG	0.9283	0.9109	0.9496	0.9298	0.8901	0.8575	0.9283	
Bagging	0.9249	0.9075	0.9461	0.9264	0.8856	0.8505	0.9249	

Overall comparative evaluation resampling models and ensemble classifiers

This comparative analysis presents a comparison of the ROC AUC scores for eight resampling methods and the baseline (no resampling) across ten classifiers, evaluating the impact of different resampling techniques on classifier performance, as shown in Table 7. In the table, the resampling methods and classifiers are sorted by their mean ROC AUC scores in descending order, which are shown in the last column and row. Additionally, in order to make the most successful combinations of resampling techniques and classifiers clearer, Fig. 2 illustrates their ROC-AUC scores for the top ten combined models, providing a visual representation of their performance.

Table 7 Comparison of sampling methods and classifiers based on ROC AUC scores.

Classifier/Sampling	IHT	SMOTE-ENN	RENN	ENN	KM-SMOTE	SMOTE-TL	ROS	SMOTE	Baseline	Mean	
ExtraTrees	0.9799	0.9899	0.9695	0.9509	0.9240	0.9363	0.9466	0.9163	0.8572	0.9412	
BalancedRF	0.9769	0.9853	0.9595	0.9486	0.9199	0.9318	0.9404	0.9139	0.8640	0.9378	
Random Forest	0.9787	0.9856	0.9586	0.9422	0.9213	0.9318	0.9385	0.9139	0.8529	0.9360	
BalancedBG	0.9756	0.9813	0.9474	0.9388	0.9123	0.9283	0.9351	0.9096	0.8560	0.9316	
Bagging	0.9711	0.9816	0.9500	0.9343	0.9139	0.9249	0.9351	0.9099	0.8312	0.9280	
LightGBM	0.9762	0.9807	0.9439	0.9231	0.9075	0.9180	0.9123	0.8963	0.8199	0.9198	
XGBoost	0.9706	0.9727	0.9394	0.9279	0.9058	0.9031	0.9039	0.8887	0.8144	0.9141	
CatBoost	0.9643	0.9672	0.9308	0.8996	0.9011	0.8965	0.8829	0.8786	0.7896	0.9012	
RUSBoost	0.8987	0.8611	0.8321	0.7719	0.8381	0.7830	0.7744	0.7802	0.7445	0.8093	
EasyEnsemble	0.8237	0.7890	0.7748	0.7643	0.7895	0.7341	0.7146	0.7167	0.7131	0.7578	
Mean	0.9516	0.9494	0.9206	0.9001	0.8934	0.8888	0.8884	0.8724	0.8143	0.8977	

Figure 2 ROC-AUC performances for the top ten combined models.

The results highlight that SMOTE-ENN generally produces the highest performance across the classifiers, with ExtraTrees achieving the highest ROC AUC score of 0.9899. IHT follows closely behind, with ExtraTrees again performing the best at 0.9799. This indicates that IHT and SMOTE-ENN are highly effective resampling methods, particularly for ExtraTrees, BalancedRF, and Random Forest. These classifiers maintain strong ROC AUC scores across all resampling methods. In contrast, RUSBoost and EasyEnsemble demonstrated significantly lower ROC AUC scores across all resampling methods, with the weakest performances observed in the SMOTE and Baseline configurations. In summary, SMOTE-ENN, IHT, and RENN methods demonstrated superior classification performance, achieving the highest ROC AUC scores when paired with tree-based ensemble classifiers including ExtraTrees, BalancedRF, and Random Forest.

On the other hand, the Baseline (no resampling) consistently shows the lowest scores across all classifiers, highlighting the importance of resampling methods. For instance, ROC AUC scores improve significantly from 0.8143 (Baseline) to 0.9516 with IHT, as shown in the mean values (see Table 7), demonstrating the effectiveness of resampling in enhancing classifier performance.

Confidence-based evaluation of high-priority B-cell epitope candidates

In this phase, the model was evaluated on the validation set using correct predictions; misclassifications were examined to identify limitations. To provide a more comprehensive evaluation of the model’s behavior, representative prediction examples with the highest confidence scores for correct and incorrect classifications are presented in Table 8.

Table 8 Top 10 most confidently correct and incorrectly classified B-cell epitopes based on model predictions.

Type	Peptide sequence	True label	Predicted label	Prediction probability	
Correct	SVATVQLP	0	0	1.00	
ECKCLLNYK	0	0	1.00	
RKAGELTLQF	0	0	1.00	
FFSSLRCMVD	0	0	1.00	
LSYFKQPVAA	0	0	1.00	
QPDLIPL	1	1	1.00	
PVMFVSRVPAKKPRK	1	1	1.00	
GKFLEGA	1	1	1.00	
TVRHRCQAIRKKPLP	1	1	1.00	
LIPLLRT	1	1	1.00	
Incorrect	KNVSPEARHPLV	1	0	1.00	
LYAPDMFEYM	1	0	1.00	
LFEDAYLLTLHAGRV	1	0	1.00	
IALWNLHGQALFLGI	1	0	1.00	
RLAKLSDGVAVLKVG	1	0	1.00	
INYWLAHKALCSEKL	0	1	1.00	
PTQAPVIQLVHAVYD	0	1	0.98	
QLFDRILL	0	1	0.96	
IFQDHPL	0	1	0.96	
YETFSKL	0	1	0.94	

Peptides such as “SVATVQLP”, “PVMFVSRVPAKKPRK”, and “TVRHRCQAIRKKPLP” were correctly classified with 100% confidence and may serve as candidate epitopes, subject to experimental validation. In contrast, peptides like “KNVSPEARHPLV” and “LFEDAYLLTLHAGRV” were misclassified with high confidence, highlighting potential model limitations and complexity in peptide feature representation.

Peptides that were correctly classified with high confidence represent sequences that the model consistently identifies as likely B-cell epitopes. While these peptides may hold immunogenic potential, such properties must be confirmed through biological validation. Their ability to elicit antibody responses makes them particularly valuable for immunological studies. Misclassified high-confidence peptides reveal model limitations and potential complexities in the sequence data, such as structural features or mutations. These observations highlight which peptide types the model handles well or struggles with, and provide direction for model refinement and future experimental studies. Ultimately, biological validation remains essential to confirm the functional relevance of all predicted candidates.

Statistical significance tests

To confirm that the observed improvements in classification performance were not due to random variation or changes in data structure, various paired statistical tests were applied. Specifically, we conducted three widely accepted statistical tests based on the ROC AUC scores obtained from five-fold cross-validation. These included the paired t-test, the Wilcoxon signed-rank test, and the permutation test. The paired t-test evaluates whether there is a statistically significant difference between the mean performance scores of two models. The Wilcoxon signed-rank test, a non-parametric method, assesses median differences between paired distributions. The permutation test estimates the p-value by repeatedly permuting the labels of the data to determine whether the observed differences could have occurred by chance. For example, when comparing the Baseline and SMOTE-ENN resampling methods using the Random Forest classifier, statistically significant differences were observed in all tests. In general, performance improvements obtained through resampling methods were found to be significantly better than those of the baseline scenario. The average ROC AUC difference between the baseline and resampling techniques was calculated as 0.058290. The paired t-test yielded a p-value of 0.000086, the Wilcoxon test yielded 0.015620, and the permutation test yielded 0.015050. Since all p-values are below the 0.05 threshold, the differences can be considered statistically significant. These findings confirm that the performance gains achieved through resampling methods are not merely due to random variations, but rather reflect statistically significant model improvements.

Discussion

The findings of this study underscore the significant potential of integrating resampling strategies and ensemble learning models for the accurate prediction of B-cell epitopes for vaccine design in SARS-CoV and SARS-CoV-2. The combination of various resampling techniques, such as over-sampling, under-sampling, and ensemble balancing, has proven essential in addressing the class imbalance issues inherent in epitope prediction tasks (Wang et al., 2020; Thomas et al., 2022). By enhancing the representation of both positive and negative epitope samples, these methods contribute to the model’s improved performance, ensuring that predictions are not biased toward the majority class (Jain et al., 2021; Noumi et al., 2021; Ozger & Cihan, 2022). A key takeaway is the substantial improvement in model performance when advanced resampling techniques, such as IHT and SMOTE-ENN, were applied. These methods effectively addressed the class imbalance present in the dataset, resulting in a significant increase in ROC AUC scores (Gurcan & Soylu, 2024b). Notably, IHT and SMOTE-ENN consistently outperformed baseline methods, with IHT achieving the highest average ROC AUC score, while SMOTE-ENN enhanced ExtraTrees’ performance to an impressive ROC AUC of 0.9899 (Kraiem, Sánchez-Hernández & Moreno-García, 2021). These results underscore the effectiveness of data rebalancing strategies in enhancing the predictive performance of ensemble classifiers (Bukhari et al., 2022; Grewal, Hegde & Yanow, 2024; Gurcan & Soylu, 2024b, 2024a).

The use of ensemble models, incorporating bagging, boosting, and balancing methods, has demonstrated superior predictive accuracy compared to individual models (Haixiang et al., 2017; Gurcan & Soylu, 2024a). The ability to predict epitopes with high accuracy is crucial for the development of vaccines and therapeutic strategies targeting SARS-CoV and SARS-CoV-2, as it directly impacts the identification of promising targets for immune response modulation (Jain et al., 2021; Bukhari et al., 2022; Palma, 2023). Moreover, the methodology developed in this study, which incorporates both structural and chemical protein properties alongside sequence-based features, provides a more comprehensive approach to epitope prediction (Jain et al., 2021; Noumi et al., 2021; Thomas et al., 2022). This multi-dimensional analysis allows for a deeper understanding of the factors influencing epitope-antibody interactions, offering valuable insights into the immunological properties of the SARS-CoV and SARS-CoV-2 viruses (Sanchez-Trincado, Gomez-Perosanz & Reche, 2017; Caoili, 2022; Ozger & Cihan, 2022). These findings can contribute to more targeted vaccine designs and therapeutic interventions, particularly in the context of emerging variants and their ability to evade immune detection (Wang et al., 2020; Caoili, 2022; Zheng, Liang & Zhang, 2023). Experimental validation is crucial to confirm the biological relevance of predicted epitopes. Without lab-based validation, the predictions remain theoretical and may not accurately reflect immune system behavior (Bukhari et al., 2022; Palma, 2023). Future research should focus on validating these predictions through experiments, ensuring their applicability in vaccine development and therapeutic strategies (Toride King & Brooks, 2018; Xu & Kulp, 2019; Ozger & Cihan, 2022).

Despite promising results, several limitations should be addressed in future research. The study relies on available datasets, which may not capture the full diversity of epitopes across different strains or populations (Jain et al., 2021; Noumi et al., 2021). Data preprocessing techniques like feature selection and normalization might not fully optimize model performance, as their effectiveness depends on the dataset quality (Thomas et al., 2022; Gurcan & Soylu, 2024b). Although the combination of powerful resampling and classification techniques such as SMOTE-ENN and ExtraTrees yielded high ROC AUC scores, it may also pose a potential risk of overfitting. Acknowledging this concern, our study employed seven different evaluation metrics (ROC AUC, PR AUC, Accuracy, Precision, Recall, F1-score, and MCC) during performance assessment, with overall evaluations primarily based on the ROC AUC score. Therefore, the high performance results should be interpreted comprehensively in light of all evaluation metrics and should not be solely limited to internal validation. To better assess the generalizability of the model, future studies should include external validation using different and independent datasets. Such efforts will enhance the reliability of the model in real-world applications. In addition, exploring advanced machine learning models, such as deep learning, could further enhance prediction accuracy, complementing the strong performance of ensemble methods (Bukhari et al., 2022; Grewal, Hegde & Yanow, 2024). The study’s contributions go beyond epitope prediction, setting a precedent for future research by integrating resampling and ensemble techniques to develop more accurate and generalizable models (Sanchez-Trincado, Gomez-Perosanz & Reche, 2017; Jain et al., 2021; Kraiem, Sánchez-Hernández & Moreno-García, 2021; Thomas et al., 2022). In conclusion, this study emphasizes the value of resampling and ensemble learning in improving epitope prediction for SARS-CoV and SARS-CoV-2, advancing our understanding of viral immunology and supporting future vaccine development efforts against these and similar diseases.

Conclusion

This study explores the use of ensemble models with resampling strategies to enhance antibody valence recognition, focusing on linear B-cell epitope prediction for vaccine design targeting SARS-CoV and SARS-CoV-2. Addressing challenges in imbalanced datasets, it provides a detailed evaluation of various resampling methods and ensemble classifiers, with the evaluation focusing on their performance across multiple metrics, particularly ROC AUC scores. By combining these methods, the analysis highlights strategies for improving predictive capabilities in immunoinformatics.

The findings highlight that advanced resampling techniques, such as IHT and SMOTE-ENN, consistently outperform baseline methods by significantly improving classifier performance. Notably, IHT achieved the highest average ROC AUC score of 0.9516, demonstrating its effectiveness in enhancing the predictive capabilities of classifiers. SMOTE-ENN achieves the best performance, with ExtraTrees reaching the highest ROC AUC score of 0.9899. Among the ensemble classifiers, ExtraTrees, BalancedRF, and Random Forest consistently ranked as the top-performing models, achieving robust scores across all metrics, especially when paired with effective resampling methods. In contrast, the baseline (no resampling) method consistently showed the lowest performance, underscoring the importance of applying resampling techniques to address class imbalance. For example, the ROC AUC scores for ExtraTrees increased from 0.8572 with the baseline to 0.9899 with SMOTE-ENN, demonstrating the transformative impact of appropriate resampling.

This study contributes by exploring rebalanced ensemble models to enhance antibody valence recognition, particularly for linear B-cell epitope prediction in vaccine design targeting SARS-CoV and SARS-CoV-2. It addresses class imbalance through various resampling techniques and ensemble classifiers, focusing on key performance metrics, especially ROC AUC scores. The analysis reveals effective strategies to improve predictive capabilities in immunoinformatics, offering valuable insights for future research. Future work should explore integrating deep learning with resampling methods to enhance accuracy, as well as incorporating GANs for synthetic data generation. Expanding datasets and epitope prediction tasks will help generalize findings and improve model robustness.

Supplemental Information

Supplemental Information 1 Source codes of the models.

Additional Information and Declarations

Competing Interests

The author declares that they have no competing interests.

Author Contributions

Fatih Gurcan conceived and designed the experiments, performed the experiments, analyzed the data, performed the computation work, prepared figures and/or tables, authored or reviewed drafts of the article, and approved the final draft.

Data Availability

The following information was supplied regarding data availability:

The raw data is available at Noumi et al. (2021) (DOI: 10.2197/IPSJJIP.29.321) and Kaggle: https://www.kaggle.com/datasets/futurecorporation/epitope-prediction.

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
