# Peer review of "Linear B-cell epitope prediction for SARS and COVID-19 vaccine design: Integrating balanced ensemble learning models and resampling strategies"

_PeerJ Computer Science, doi:10.7717/peerj-cs.2970_

## Round 0.1 · original submission · Major Revisions

The reviewers have substantial concerns about this manuscript. The authors should provide point-to-point responses to address all the concerns and provide a revised manuscript with the revised parts being marked in different color.

Reviewer 1 ·

Basic reporting

Gurcan’s study presents a novel integration of resampling techniques and ensemble learning for epitope prediction, which is commendable. I would endorse the contribution of this manuscript in computational immunology and vaccine design.

Experimental design

1) Experimental setup and performance evaluation section, the manuscript does not explain how hyperparameters were chosen for ensemble models. I strongly suggest adding a section on hyperparameter optimization techniques (e.g., grid search, Bayesian optimization), which would improve the study’s reproducibility.
2) Lines 168–172: The study expands feature sets, but it does not discuss whether any feature selection methods (e.g., Principal Component Analysis, Recursive Feature Elimination) were used to determine the most important variables. Adding this discussion would clarify the model’s robustness.
3) Line 28: "SMOTE-ENN and Extra Trees yielded the best performance, achieving an ROC AUC score of 0.9844." This is a strong claim. Include a brief discussion on potential overfitting risks and the importance of external validation.
4) Section 3.5. The discussion focuses on different resampling strategies but does not compare against non-resampling methods like deep learning classifiers. Consider adding a comparison with other standard classifiers without resampling.

Validity of the findings

1) Lines 135–137: The dataset link should be explicitly mentioned in the text rather than just referencing Kaggle.

Additional comments

1) Lines 90–92: "The study leverages robust datasets obtained from reliable sources such as IEDB and UniProt..." The word "leverages" is unnecessarily complex; replacing it with "uses" improves readability.
2) Line 49: "These antibodies, as part of the immune defense, play a vital role in combating infections." Suggested revision: "As part of the immune defense, these antibodies play a crucial role in combating infections."

Reviewer 2 ·

Basic reporting

no comment

Experimental design

no comment

Validity of the findings

no comment

Additional comments

This study explores the use of data balancing methods and ensemble learning for predicting B-cell epitopes, aiming to improve model accuracy when dealing with uneven data. The authors apply methods such as SMOTE-ENN and IHT and report a clear increase in ROC AUC scores. However, there are several important issues with the study’s methods and explanations. The paper does not include tests (e.g., McNemar’s test, paired t-test) to confirm that the reported improvements are not due to chance or changes in data structure rather than real model improvements. The feature engineering step focuses mainly on sequence length and position but does not explain why these were chosen instead of other meaningful biological factors, such as chemical properties or protein structure. The authors also remove sequence identifiers (protein_seq, peptide_seq, and parent_protein_id) without checking whether these contain useful information, which may weaken the model. Additionally, the study relies too much on ROC AUC to measure success, without using other measures such as PR AUC or MCC, which are better suited for uneven data.


1. In the MATERIALS AND METHODS, this study exhibits issues in experimental design and data processing regarding the application of resampling and ensemble learning, which affect the reliability of the results and their biological significance. Firstly, resampling methods such as SMOTE, IHT, and SMOTE-ENN directly alter the data distribution, yet the study does not analyze the specific changes in amino acid sequences and physicochemical properties before and after resampling. Artificially balanced data may introduce biologically unrealistic samples, causing the model to learn features that lack practical value for B-cell epitope prediction. Secondly, the study only compares resampling-based models with a "no resampling" baseline but does not incorporate alternative strategies such as cost-sensitive learning, which can handle class imbalance without modifying the data distribution. This makes it unclear whether resampling truly enhances model performance or simply alters the dataset structure. Furthermore, performance evaluation relies primarily on ROC AUC, which, in highly imbalanced datasets, may mask the model’s ability to correctly identify minority class instances. In contrast, PR AUC or MCC would provide a more accurate assessment of classification performance, reducing the risk of overestimating the model’s generalizability.
2. The study introduces sequence length and positional attributes as engineered features, but it does not provide a clear rationale for why these specific features were selected or how they directly contribute to B-cell epitope prediction. Given that epitope recognition is influenced by structural and physicochemical properties, what evidence demonstrates that these added features improve predictive performance rather than merely increasing the number of variables? Additionally, the study removes sequence identifiers (protein_seq, peptide_seq, and parent_protein_id) without assessing whether they contain useful sequence-derived information. Was any analysis conducted to determine the impact of this removal on model performance, or was this decision made purely for data formatting purposes?
3. The results and discussion section states that applying resampling strategies, particularly SMOTE-ENN and IHT, improved model performance. However, no clear statistical analysis is provided to quantify the reliability of this improvement. Does the current experiment distinguish between an actual enhancement in the model’s generalization ability and a mere alteration in data distribution that makes it easier for the classifier to achieve higher scores on the training set? If the observed increase in ROC AUC is not supported by paired statistical tests, such as McNemar’s test, paired t-test, or Wilcoxon signed-rank test, it remains unclear whether the performance gains stem from data adjustments or represent a statistically meaningful improvement.

Reviewer 3 ·

Basic reporting

no comment

Experimental design

no comment

Validity of the findings

no comment

Additional comments

The authors propose a resampling-ensemble hybrid approach to address the challenge of class imbalance in B-cell epitope prediction. The article is written in clear and professional English and the logic is understandable, supported by well-referenced and relevant literature. The reviewer supports the manuscript for publication until the following issues are addressed:
1. Add a comparison with existing similar studies, including but not limited to the following:
o PMID: 36016459 PMCID: PMC9416013 DOI: 10.3390/v14081837
o PMID: 37776561 PMCID: PMC10541793 DOI: 10.1093/database/baad065
o PMID: 33486372 PMCID: PMC7737509 DOI: 10.1016/j.jiph.2020.12.006
o PMID: 34811480 PMCID: PMC8608966 DOI: 10.1038/s42003-021-02835-2
The manuscript should sufficiently distinguish how their integration approach advances beyond existing studies. For example, after applying the resampling-ensemble method, are any new epitopes being predicted? Are there any epitopes with higher accuracy or confidence? Explain why people should follow the authors' hybrid approach instead of using the pre-existing SARS-CoV-2 B-cell epitope prediction results. Providing a more direct and thorough comparison will strengthen the contribution and significance of their work.
2. Dataset: The authors leverage IEDB and UniProt, which are appropriate. It would be better to validate their methodology on additional datasets, especially those available for SARS-CoV-2 epitopes.
3. Provide a better presentation of B-cell epitopes. The authors state the advances of “integrating resampling models and balancing ensemble classifiers in improving the accuracy of B-cell epitope prediction and antibody recognition for SARS and COVID-19 infections.” Besides the statistical tables, maybe a figure showing the top 5 or top 10 predicted epitopes revealed with/without their integration approach will help readers gain a better perceptual impression.
4. Line 72-74: “Traditional methods, including X-ray crystallography and nuclear magnetic resonance (NMR) spectroscopy, are widely employed to analyze the three-dimensional structures of antigen-antibody interactions.” This sentence should include cryo-EM as well.
5. Line 75-77: “While these techniques provide highly accurate and reliable results, they are costly and time-intensive, limiting their feasibility for widespread application.” This sentence should also consider AF2/AF3, then be rewritten.
6. Line 107-109: “New descriptor features were added to the empirical dataset using protein-peptide and epitope-antibody interaction information obtained from reliable data sources such as IEDB and UniProt.” Did any similar work also use data sources from IEDB and UniProt?
7. Line 110-112: “The methodology enables more comprehensive and multidimensional analyses by considering not only target amino acid sequences but also structural and chemical protein properties.” Can the authors comment on the performance compared to AF3? This question is not mandatory and the authors may decline to answer if they prefer.

---

## Round 0.2 · Major Revisions

There are some remaining major concerns that need to be addressed.

Reviewer 1 ·

Basic reporting

no comment

Experimental design

no comment

Validity of the findings

no comment

Additional comments

The author has successfully addressed my concerns, and I recommend accepting this manuscript in its current version.

Reviewer 2 ·

Basic reporting

The manuscript presents a promising machine learning framework for B-cell epitope prediction. However, concerns regarding potential overfitting, inappropriate use of the test set for candidate selection, and biological overinterpretation of predicted epitopes need to be addressed. Additionally, external validation is necessary to strengthen the generalizability of the results. I recommend major revision, focusing on correcting methodological issues.

In the current manuscript, the statement "To interpret the model's predictive power, peptides with the highest confidence in both correct and incorrect classifications were identified, highlighting potential epitope candidates" reveals a notable logical flaw. Scientifically, while high-confidence correct classifications are indeed appropriate for identifying promising epitope candidates, high-confidence misclassified peptides should not be treated the same way. Misclassified samples, even if predicted with high confidence, indicate areas where the model fails and thus should serve as warning signals rather than candidate selections.
In the EXPERIMENTAL RESULTS, In the "Confidence-based evaluation of high-priority B-cell epitope candidates" section, the authors select the top 10 most confidently correct and incorrect predictions from the test set for further analysis. However, using the test set in this way is not fully appropriate from a methodological standpoint. In machine learning practice, the test set is intended solely for the final evaluation of model performance and should not be used for sample selection or additional interpretation. Selecting candidate peptides based on test set outcomes could lead to biased conclusions and compromises the independence of the performance assessment. It would be more appropriate to perform candidate selection using the training or validation data, and to reserve the test set for independent evaluation only.
In the discussion section, the authors suggest that peptides predicted with high confidence "may possess high immunogenic potential" and could be "strong vaccine candidates pending experimental validation." However, this conclusion extends beyond what the available data can substantiate. Machine learning-based epitope prediction, even with high-confidence scores, does not directly equate to confirmed immunogenicity without biological validation.

Experimental design

no comment

Validity of the findings

no comment

Reviewer 3 ·

Basic reporting

N/A

Experimental design

N/A

Validity of the findings

N/A

Additional comments

The authors have addressed all the issues in this revised manuscript. The reviewer supports its publication.

---

## Round 0.3 · accepted · Accept

All concerns have been addressed and I recommend accepting this manuscript.

Reviewer 2 ·

Basic reporting

The author has responded to my concerns and addressed some of my suggestions. I have no additional questions.

Experimental design

The author has responded to my concerns and addressed some of my suggestions. I have no additional questions.

Validity of the findings

The author has responded to my concerns and addressed some of my suggestions. I have no additional questions.